# Research Framework Built Natural-Based Solutions (NBSs) as Green Hotels

Taeuk Kim [1] and Sunmi Yun [2],*

1   Department of Hotel & Restaurant Management, Kyonggi University, Seoul 03746, Korea;
    teokim1305@naver.com
2   Department of Hospitality and Tourism Management, Sejong University, Seoul 05006, Korea
*   Correspondence: sunmiyun80@gmail.com; Tel.: +82-10-4055-2066

**Abstract:** In this study, value-belief-norm (VBN) theory and the social exchange theory (SET) were applied to predict hotel customers' pro-environmental responsibility behavior intention (PRBI) for the characteristics of NBSs in green hotels—specifically, to investigate the relationship between NBSs as green hotel and PRBI, and to test its mediating effect on pro-environmental perceived (PPV), pro-environmental perceived belief (PPVBE), personal pro-environmental norms (PPN), attitude toward environmental behavior (ATEB), mental health (MH), well-being (WB), and satisfaction (SA) and the moderating effect of locations (urban, rural) among these variables toward pro-environmental responsibility behavior intention (PRBI). Data were collected using a survey of 440 customers who had visited green hotels in the Republic of Korea within the last 12 months. We used to test the research hypotheses by structural equation modeling (SEM). The findings generally supported the hypothesized associations between variables within our proposed theoretical framework and confirmed the moderating effect of location. The study's results have important theoretical and practical implications for the environment. We investigated the relationship between the characteristics of NBSs and PRBI of green hotels, and we investigated the relationship between psychological state, attitude, and behavior of green hotel customers by applying variables suitable for ART, SET, and VBN. In addition, we verified the moderating effect of customers' green behavior and attitudes toward green hotels located in urban and rural areas. Moreover, these findings herein may encourage green hotels to participate in preventing environmental problems. It provides primary data on customers' perception of ecofriendliness in establishing corporate management strategies.

**Keywords:** natural-based solutions (NBSs); pro-environmental value (PPV); pro-environmental perceives belief (PPVB); personal pro-environmental norms (PPN); attitude toward environmental behavior (ATEB); mental health (MH); well-being (WB); satisfaction (SA); pro-environmental responsibility behavior intention (PRBI); green hotels

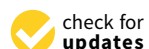



## 1. Introduction

As COVID-19 spread rapidly around the world and the number of confirmed cases and deaths continued to increase, the World Health Organization (WHO) declared the disease a pandemic on 11 March 2020 [1], the third pandemic in history after Hong Kong flu in 1968 and novel swine-origin influenza in 2009. For this reason, it has been hit hard in various fields such as politics, economy, and culture [1].

The continued effects of COVID-19 are engendering a strong sense of crisis in terms of the environment and the impact of changes to consumer life [2]. The world is not only facing the COVID-19 pandemic but is also struggling with the increase in waste mainly from plastics affecting both society and healthcare sectors [2–4]. For instance, there is unintentionally and rapidly generated plastic waste from single-use plastics such as facial masks, hand sanitizer bottles, surgical gloves, takeout food packaging, food and polyethylene goods packaging, protective medical suits, and medical test kits [4]. This

is because of disposable products and materials for health and safety reasons, which has worsened the plastic pollution problem [5]. According to 'The Global Risks Report 2020' published by the World Economic Forum, environmental problems present the most significant risk today.

As such, interest in global environmental problems has increased due to the impact of COVID-19, and in response, natural-based solutions (NBSs) have emerged as an essential global phenomenon [2]. The concept of NBSs is a growing global phenomenon that can help societies address various social and environmental challenges sustainably [6,7]. Due to sustainable characteristics and the pursuit of social and environmental benefits, NBSs have attracted attention in various contexts in recent years [6]. NBSs are presently being applied to the design of residential areas, workplaces, buildings, and urban spaces to enhance the quality of human life [6–9]. Hotels are increasing their NBSs endeavors to create green spaces inside the building and protect the natural outdoor environment [8]. This creates what NBSs call green building that result in eco-environmental physical environment that influences human responses and behaviors [8,10]. Thus, hotel green spaces, together with pro-environmental physical environments, often play a crucial role in positively affecting the well-being (WB) and mental health (MH) of the customer to encourage continued retention [2,8,9].

This study focused on Value-Belief-Norm (VBN) theory, attitude toward sustainability, and psychological health as NBSs to explicate customer pro-environmental responsible behavior intention processes. We took this approach because social psychologists suggest that values, beliefs, and norms play a principal role in motivating pro-environmental behaviors [10–15]. Thus, consumers' personal characteristics such as social factors (i.e., participation in society, normative influence, and information influence), values, and personal emotions can influence patronage intention. In particular, social factors play a critical role in environmental problems [16].

Second, it was utilized as it plays an important role in understanding NBSs for ecofriendliness and customer attitudes toward the sustainable natural environment and ecofriendly behavioral intentions (responsible behavioral intentions for the environment) [17]. According to SET, the structure of society is formed through the interchange of tangible and intangible valuable things [18,19]. For example, tangible values might refer to green spaces of ecofriendly hotels, and intangible values might refer to actions that help ecofriendly hotels, such as values, beliefs, norms, and attitudes toward the environment [17]. When these tangible and intangible things are exchanged in social phenomena, they react conditionally in proportion to their relative values [20]. Applying SET to ecofriendly hotel companies can be an interchange between NBS of ecofriendly hotels and customers using these hotels.

Third, according to attention restoration theory (ART), NBSs of eco-friendly hotels play an essential role in inducing satisfaction and eco-friendly behavioral intentions with a positive effect on customer MH and WB [8,9,21–25]. ART is a concept of the psychological recovery effect natural spaces provide to humans [20]. In environmental psychology, natural space research on individual happiness and recovery has been conducted, revealing that nature provides positive emotional outcomes such as improving MH and restoring attention [26]. For instance, Korea is currently considering possible travel (visiting ecofriendly hotels) to recover from fatigue caused by COVID-19 and daily life. Because overseas travel has been restricted due to the COVID-19 pandemic, hotel stays are enjoying popularity as an alternative travel option; moreover, people prefer green hotels when choosing where to stay [27], when you feel negative emotions in your daily life, your desire to feel happiness through leisure and travel activities increases [28].

To summarize the above, the first purpose of this study is to verify the impact relationship on PRBI of customers who have stayed at green hotels as NBSs. The second purpose is to verify the mediating effect by applying VBN to explain the customer's ecofriendly behavior and SET to explain the customer has perceived costs and benefits. Finally, to examine changes in perceptions and behaviors at green hotels in an urban and rural locations. Therefore, this study aims to provide academic and practical implications for the influence

relationship of NBS as green hotels on customers' PRBI through various variables applied in the theories of VBN, ART, and SET.

## 2. Conceptual Framework

### 2.1. Value-Belief-Norm Theory

VBN is a theory that explains people's ecofriendly behavior as three factors—value, belief, and norm—and is an extension of the norm activation model [13]. VBN theory is most widely used along with the theory of planned behavior when analyzing factors related to environmental behavior [15]. VBN theory was developed with a focus on environmentalism [15]. It integrates the perspective of value theory [14], the norm activation model of [29], and the new environmental paradigm of [13,30]. Thus, VBN theory identifies ecofriendly behavior by organizing the factors influencing ecofriendly behavior into values, new paradigms, perception of results, ascribed responsibility, and personal norms. In particular, personal norms (such as being willing to pay premium prices for ecofriendly food [31]) are noted in previous studies, such as [32], which focus on ecofriendly organic wine purchase intention. It was focused that personal norms having a positive effect on ecofriendly travel [33].

In addition, [15] proposes an action based on norms according to [29]'s theory of activation of norms. This process refers to the belief that an individual accepts a specific value, and if that value is threatened, actions are taken to help alleviate the threat and restore its value to activate personal norms. Moreover, the VBN theory explains how an individual's moral obligation leads to action. The value perceived by an individual leads to a norm that forms a belief and determines an individual's behavior [13]. The mindset factor most related to the actual behavior of VBN theory is a personal norm, which can be defined as creating a moral obligation to practice as an action and a willingness to engage in environmentally friendly behavior. Thus, personal norms are essential variables that can predict the behavior of ecofriendly consumers in various environments [13].

In this context, previous studies in the tourism industry using VBN are described as follows. In a study by [34] targeting users of national parks in India, analysis of the impact of the relationship with ecofriendly behavior of users applying the VBN model verified that ecofriendly values, attitudes, and norms are important variables in predicting the ecofriendly behavior of nature-based tourists. In addition, in a study by [35] targeting restaurant users with experience in using delivery food services, the VBN model was applied, confirming the significant positive effect in intention to use, word of mouth, and additional cost payment. In a study conducted by [36] on Vietnamese travelers, the analysis of ecofriendly behavioral intentions on Vietnamese agricultural tourism showed a difference in ecofriendly behavior between Vietnamese locals and overseas travelers.

These studies verified that these characteristics positively affect the ecofriendly behavior of users and travelers. In this context, studies using the concept of NBSs are gradually increasing about ecofriendly behavior, a study by NBSs applied to the service industry [3,37]. The successful implementation of NBS in the hotel industry can improve both MH and WB awareness of customers and employees [9]. Green spaces, green surfaces, green items, and natural environments are all part of environmentally friendly hotels and buildings whose service performance or characteristics are the main feature of product quality evaluation [38–40], which can provide high psychological and emotional value to customers.

Specifically, previous studies in the tourism industry applying VBN are as follows. In a study by [41] targeting customers who chose ecofriendly hotels (linen reuse program, recycling gray water from sinks and shows, low-flow water fixtures), a significant influence on the relationship between attitudes toward ecofriendly hotels and eco-friendly responsibility behavior was verified. In addition, by verifying the behavioral results in a study by [42] that applied VBN, it was confirmed that the attitudes of those who demonstrated eco-friendly values increased with a commitment to the environment, intention green hotels, and willingness to pay a premium price.

Here, the VBN theory can explain how the eco-friendly elements of NBSs (that is, the characteristics provided by eco-friendly hotels that build eco-friendly environments) can increase perceived value for customers. Therefore, based on the prior studies, the hypothesis is established that there is a relationship between eco-friendly values, beliefs, and norms in eco-friendly hotels.

**Hypothesis 1 (H1).** *Green hotels as NBSs significant effect on pro-environmental perceived value.*

**Hypothesis 2 (H2).** *Pro-environmental perceived value significantly effects on pro-environmental perceived belief.*

**Hypothesis 3 (H3).** *Pro-environmental perceived belief has a significant effect on personal norms.*

**Hypothesis 4 (H4).** *Satisfaction has a significant effect on pro-environmental responsible behavior intention.*

### 2.2. Social Exchange Theory

SET is a general sociological theory related to understanding resource exchange between individuals and groups in interaction situations [43]. In the process of exchange, members of society rely on each other for the consequences of what they value. Hence, they try to obtain more positive values and reduce negative values, assuming that all individuals participating in the interaction process are based on subjective cost–benefit analysis and alternative comparison. In this process, individuals continue to interact when they feel more benefits are generated [44]. According to social exchange, human behavior begins in exchanging costs and rewards with others. In the social exchange theory developed by [18], it is said that when humans receive benefits such as compensation from the other party of exchange, they form an exchange relationship that makes them feel obligated to repay it someday in the future. In this context, SET-applied tourism suggests a sense of obligation among consumers to pay back in the future when an 'exchange relationship' is established through tourism activities when compensation or benefits are obtained from the other party. Based on the tangible and intangible resources of tourism effects through tourism experiences, tourists and residents form an attitude toward tourism by evaluating and comparing the benefits and costs they earned through the exchange process [45]. In other words, healthy life is affected by the surrounding social and physical environment, especially when people and the natural environment form a complementary relationship; accordingly, the environment can improve people's mental and physical health [6].

Lovaglia explained social exchange relations by dividing them into economic and social exchanges [17]. It was explained that social exchange theory is a relationship that has a sense of obligation to pay someday when receiving benefits such as compensation from the other party, and that economic exchange is based on the response method of the object to be exchanged, such as the transaction of goods. Customers who use the hotel choose a situation where they can benefit a lot by maximizing the benefits of using the hotel and minimizing the cost. In this context, when customers stay at green hotels, they can learn how to do ecofriendly behavior, enjoy fresh and healthy food, and benefit from functional and emotional benefits such as making you feel good by others, while they perceive prices relatively high and pay for financial and nonmonetary costs.

Moreover, SET predicts that a person will leave a relationship when they perceive the costs of the relationship outweigh the benefits [46]. This framework is considered suitable for examining residents' perceptions of tourism [47,48]. Residents evaluate tourism in terms of social exchange (expected benefits or costs in return for the services they supply). Perceived high costs may stimulate negative attitudes from the host— for example, though, over-exploitation [47]. SET has not been applied in the content of green hotels, while some studies suggest that customers perceive functional and emotional value during their stay at green hotels [49], although the price was the only cost item included in the value measurement scale [50]. Although it is unclear what green hotel customers will sacrifice,



many studies related to green product consumption or daily eco-friendly activities suggest high monetary and nonmonetary costs, and risks associated with making lifestyle changes hinder customers from going green [51].

Examining prior research in the tourism industry reveals a significant effect of ecotourism on mental health, satisfaction, and behavior. One study [9] found positive effects from a survey of ecofriendly surfaces hotel users in Korea. NBSs were explained in the order of naturally ecofriendly, ecofriendly surfaces, ecofriendly spaces, and ecofriendly items. After analyzing the impact of user's revisit intention by applying the expanded theory of planned behavior on customers who attended a green hotel that had received a Leadership in Energy and Environmental Design (LEED) award, [51] a significant positive influence on revisit intention was revealed according to attitude, subjective norms, and perceived behavioral control. A further study by [52] surveyed customers using ecofriendly hotels in Malaysia and found that the values and emotions derived from the physical environment in green hotels positively affected customers' attitudes and perceived behavioral control. Therefore, the following hypothesis was established. In addition, in a study focusing on green hotels conducted by [53], significant partial mediating effects were confirmed after verifying the mediating effects of MH, emotional WB, and green brand attitude in the relationship between NBSs and green brand evangelism. Therefore, the following hypothesis was established based on previous studies. A study by [54] confirmed that the perceived value of eco-friendly hotels has a significant positive effect on customers' attitudes toward green hotels. Therefore, the following hypothesis was established based on the prior studies.

**Hypothesis 5 (H5).** *Green hotels as NBSs have a significant effect on attitude toward environmental behavior.*

**Hypothesis 6 (H6).** *Attitude toward environmental behavior has a significant effect on satisfaction.*

**Hypothesis 7 (H7).** *Attitude toward environmental behavior has a significant effect on proenvironmental responsible behavior intention.*

*2.3. Green Hotels as Natural-Based Solutions*

"NBSs are acts that are inspired by, supported by, or imitated from nature" and are intended to handle a variety of environmental concerns in a cost-effective and adaptable manner while also bringing economic, social, and environmental advantages [6,8,9]. NBSs are clearly described in the final report of the Horizon 2020 Expert Group [55]. The definition of NBS is not agreed upon worldwide but is recognized and used as a helpful approach in various fields. NBSs cover all natural-based approaches, such as ecosystem-based adaptation and ecosystem-based mitigation [56]. The International Union for Conservation of Nature (IUCN) defines NBSs as a measure for protecting, continuing management, and restoring natural or transformed ecosystems that provide human well-being and biodiversity benefits [56].

NBSs are characterized by their ability to solve various social problems simultaneously as providing natural benefits. They have significant advantages in improving resilience and increasing biodiversity and sustainability in the face of climate change. Because NBSs are compatible with the human environment, they are considered an essential solution for human life and activity over time [57]. Moreover, since 2015, NBSs have been considered an optimized solution that is resilient, adaptable, resource-efficient, and locally adjustable for improving and maximizing the quality of life for urban residents [58]. Ultimately, NBSs contribute to achieving sustainable development goals such as long-term food security, climate change, water security, human health, disaster risk management, and social and economic development [59]. The OECD's approach to implementing NBSs emphasizes better recovery in overcoming the impact of COVID-19. In terms of human health, ecosystem services are essential to human economic activities, and health, deforestation, and land use

are all related to the spread of diseases [2]. Therefore, the successful implementation of NBSs to improve MH and WB awareness of both customers and employees is also very important in the hotel industry [2]. Emphasizing the importance of NBSs, psychological health and WB are also subject to study by various researchers in the tourism industry [9,53]. In terms of NBSs related to green infrastructure, green spaces, green surfaces, green items, and natural environments are essential for environmentally friendly hotels and buildings where service performance and characteristics are the main features of product quality evaluation [38–40]. The ecofriendly physical environments of hotels that accept the aspect of NBSs can have very positive results for MH and WB beyond overall customer satisfaction and satisfaction with their stay.

People's psychological aspects and attitudes are highly correlated with whether hotel users can easily access ecofriendly spaces (rivers, lakes, mountains), and whether various nature-friendly spaces are available. In a survey of hotel users [6], verifying a significant influence on ecofriendly space and psychological elasticity, customer attitudes, and behavioral intentions that explain ecofriendly accessibility, availability, and diversity, it was found that attitude has no significant effect on behavioral intention. In addition, in a study by [2] that targeted hotel customers, the specific role of the NBSs-based hotel ecofriendly physical environment was identified as a significant influencing relationship on WB, MH, and satisfaction. Therefore, NBSs play an important role in people's psychological well-being, and the following hypotheses were established by requiring in-depth and diverse studies on the benefits and effects of NBSs.

**Hypothesis 8 (H8).** *Green hotels as NBSs have a significant effect on mental health.*

**Hypothesis 9 (H9).** *Green hotels as NBSs have a significant effect on well-being.*

**Hypothesis 10 (H10).** *Mental health has a significant effect on satisfaction.*

*2.4. Pro-Environmental Responsible Behavior Intention*

In marketing, the concept of post-action greatly influences potential consumer decisions, such as repurchase intention and recommendation intention [60]. It can be said that after consumers form an attitude toward an object, it is their personal will and belief to express a specific future behavior, revisit intention or positive recommendation intention. In this context [61] on eco-friendly behavioral intentions, people who are both perpetrators of environmental problems and subjects of the issues to be solved participate in environmental protection, defined as a responsible environmental behavior divided into educational, economic, civic, physical, and legal and persuasive actions. Moreover, [62] defined environmentally friendly behavior as contributing to environmental conservation and conservation, and [63] stated that eco-friendly behavior is an action to reduce the negative impact on the environment. As such, many researchers define ecofriendly behavior as an ecological behavior to refrain from personal behavior and manage and preserve the natural environment and living environment to reduce and prevent environmental pollution damage [64]. As environmental awareness has increased, efforts have been made to clarify what antecedents of consumers' interests and attitudes toward the environment can lead to actual ecofriendly behavior in existing studies to solve environmental problems. Most studies have shown that environmental consciousness and ecofriendly behavior influence consumers' willingness to purchase environmental products [65]. In addition, studies have recently been conducted to emphasize the characteristics of NBSs in studies such as [9,53] applying NBS and to verify the influence of perceived value and satisfaction with ecofriendly behavior.

Eco-friendly service products increase satisfaction with customers, and it can be seen that such ecofriendly satisfaction has a high correlation with moral standards and behaviors toward ecofriendly action. Therefore, the preceding studies in the hotel industry that apply ecofriendly behavior as a result variable are as follows. A study by [8] surveyed ecofriendly

hotel customers in Vietnam and found that empirical satisfaction in ecofriendly hotels had a significant positive effect on moral norms and ecofriendly behavioral intentions. In addition, as a result of an experimental study on eco-friendly behavior [33] targeting Swiss-based four-star eco-friendly hotel customers, it was found that customers with high awareness of ecofriendly norms had high towel reuse. In other words, users with high norms show high ecofriendly behavior. In addition, in a study by [66] targeting participants in the Macau Eco-friendly Festival, it was verified that subjective norms, a subfactor of the theory of planning behavior, had a significant influence on eco-friendly values and ecofriendly behavior. As an environmental characteristic being developed by green hotels, previous studies empirically analyzed customer satisfaction and norms on eco-friendly behavioral intentions [8,33,67].

Thus far, studies on eco-friendly behavior, cost and benefits approach, and studies based on moral and normative interests have been examined. VBN theory, an extended model of the theory of planned behavior and norm activation model, has been a commonly used model in research that explains ecofriendly behavior. Therefore, this study established a model by combining the VBN, SET, and NBS models, and the following hypotheses were developed.

**Hypothesis 11 (H11).** *Well-being has a significant effect on satisfaction.*

**Hypothesis 12 (H12).** *Personal norms have a significant effect on pro-environmental responsible behavior intention.*

*2.5. The Moderating Effect on the Locations (Urban, Rural)*

While some studies argue that environmental concern is higher in cities [68], that urban residents are more concerned about the over-exploitation of natural resources [69], and that the perception of environmental problems increases as the size of place of residence increases [70], others suggest (after controlling for other sociodemographic variables) that there are no attitudinal or behavioral differences between the two types of samples [66]. A study by [71] compared urban (Madrid) and rural (Non-Madrid) residents living in Spain with an explanation of the social structure relationship of people's ecofriendly values, attitudes, and behaviors. As a result of the analysis, both urban and rural residents showed great environmental concern and low pro-environmental behavior. Urban residents have high environmental responsible values but low pro-environmental orientation. However, the more rural individuals are aware of environmental responsibilities, the more consistent they are in expressing behavioral intentions that are environmentally friendly. In the context of previous studies, examining the differences in perceptions, attitudes, and behaviors of urban and rural people in tourist destinations is considered very meaningful to figure out the structural relationship of value-belief-norms, attitudes, and well-being to predict green behavior according to the urban and rural green hotels. Therefore, the following hypothesis was established.

**Hypothesis 13 (H13).** *Location (urban, rural) plays a significant moderating role in the relationship between green hotels as NBSs and pro-environmental responsibility behavior intention.*

## 3. Methods

*3.1. Conceptual Framework*

In a situation where the COVID-19 continues for a long time, as the global environmental problem becomes more severe due to COVID-19, people are increasingly interested in global environmental problems [2–4]. Because of this, NBS has emerged as an essential issue in the hotel industry [2,6–9]. Therefore, the research framework has built NBSs into green hotels. The research framework of this study is composed of the following three main theories, as shown in Figure 1. First, this framework was centered on VBN theory, attitude towards sustainability, and psychological health as NBSs to explain the process

of pro-environmental responsible behavior intention. Second, SET was used because it plays an important role in understanding NBSs for eco-friendliness as well as customer's ATEB and PRBI. Last, NBSs of green hotels have a positive effect on customers' MH and WB and play an essential role in inducing SA and PRBI; ART was used as the framework for this study.

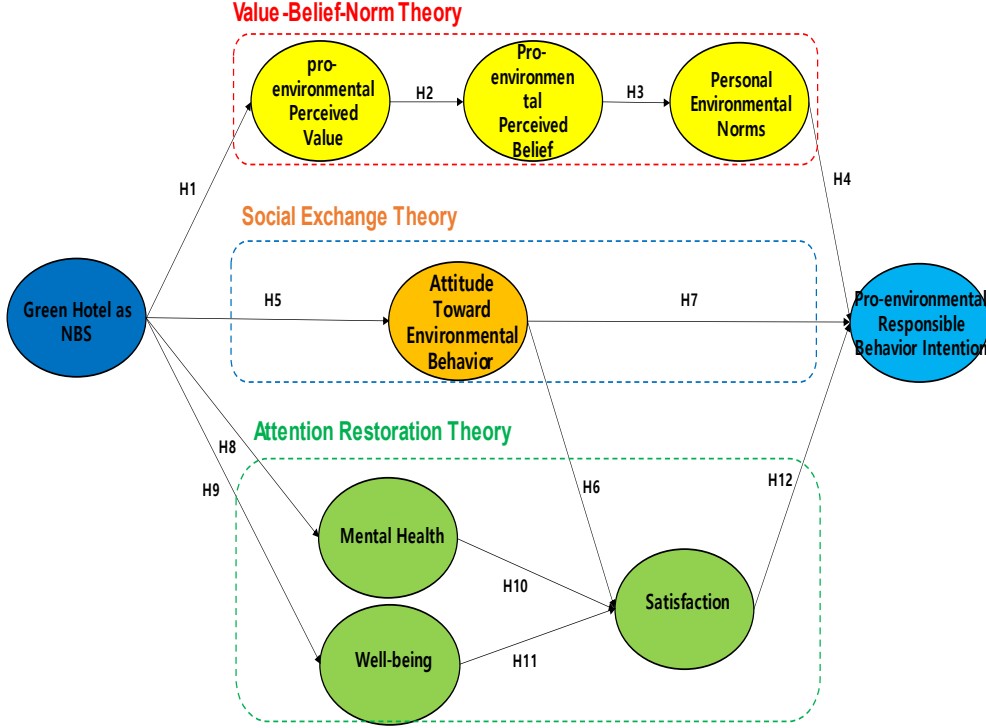

**Figure 1.** Proposal research model.

### 3.2. Measures and Questionaire Development

In order to investigate how green hotel customers' PRBI change according to NBSs, previous studies in the extant literature [2,8,9,14,38–40,53,54] provide measurement variables. The measures were modified to be adequate in the present research context. The questionnaire is presented in Appendix A. Multiple items with a five-point scale ("Strongly disagree" [1] to "Strongly agree" [5]) were used for the assessment of all study variables. In particular, four times for indoor green hotels (e.g., "I easily see green interior decorations and diverse living plants in the lobby area of green hotels") and four items for outdoor of green hotels (e.g., "The region surrounding green hotels has good and fresh quality air.]") were used. WB was evaluated with five items (e.g., "I feel healthy and happy during my stay in green hotels."). MH was measured with four items (e.g., "Staying at green hotels plays an important role in relieving my mental stress and anxiety"). In addition, we used three items for each measurement to measure PPV (e.g., "Visiting the green hotels was a pleasant experience."), PPVB (e.g., "Staying in green hotels makes you feel different from others."), PPN (e.g., "I feel that it is important to make green hotels environmentally sustainable, reducing the harm to the wider environment"), and SA (e.g., "Overall, I am satisfied with my experience at green hotels."). Moreover, four items were utilized to evaluate ATEB (e.g., "Green hotels are concerned about the environment"). Lastly, PRBI was measured with four items (e.g., "I try to participate in the activities pursued by green hotels"). The survey questionnaire containing these measures, a research description, and questions for demographic information was pre-tested by experts of two hotel practitioners and two hospitality academics. A slight modification was made by these experts based on their feedback, then we further improved and finalized the questionnaire.

### 3.3. Data Collection Process and Data Analysis

For data collection, an online research methodology used the system of an internet research company. The survey was randomly selected from the database of an internet research company, and only customers who had visited the green hotels within the last 12 months were asked to participate in the survey. A survey invitation e-mail with a research description was delivered to general hotel customers in South Korea. The survey was conducted for about three weeks, which was from 13 December 2021 to the last day of December 2021. The survey took approximately 15 min, the average time to complete. This survey was used for data analysis and the purpose of this study only. A total of 450 questionnaires were completed through this procedure. The final sample size after removing unusable responses that were used for the data analysis was 440 cases. For data analysis of this study, using SPSS 20.0 program and AMOS 20.0 program, demographic characteristics, correlation analysis, confirmatory factor analysis (CFA), structural equation modeling (SEM), and moderator effect were analyzed.

## 4. Results

### 4.1. Characteristic of Respondents

Of the 440 respondents, 292 respondents (66.4%) were female, and 148 respondents (33.6%) were male. Present marital status was highest for married people (252 respondents, 57.3%) and followed by single people (188 respondents, 42.7%). Survey participants were 30–39 years old (38.2%, 169 respondents), 40–49 years old (35.5%, 156 respondents), 50 years old or older (13.6%, 60 respondents), and 20–29 years old (12.7%, 56 respondents) appearing in order. As for the level of education, college graduates or higher (252 respondents, 57.3%), graduate school students or higher (128 respondents, 29.1%), and high school graduates (60 respondents, 13.6%) were in order, and overall, the level of education was high. The annual household income of $20,000–$39,999 (204 respondents, 46.2%) was the highest, followed by $60,000 or over (80 respondents, 18.2%), under $20,000 (80 respondents, 18.2%), and $40,000–$59,999 (76 respondents, 17.3%). In terms of the number of visits to the Green Hotel in the past year, 3~4 times (204 respondents, 46.4%), 5~6 times (84 respondents, 19.1%), 1–2 times (112 respondents, 25.2%), 10 times or more visits (7 people, 9.1%) were in that order, the purpose of visiting the green hotels was leisure (240 respondents, 72.7%), business (72 respondents, 16.4%), and other purposes (12 respondents, 10.9%). The people who visited green hotel together appeared in the order of family (164 respondents, 41.0), friends (100 respondents, 22.7%), lover (72 respondents, 14.5%), alone (72 respondents, 14.5%), and colleague (32 respondents, 7.3%). The respondents who visited the green hotel located were 168 respondents (42.7%) in the urban areas and 252 respondents (57.3%) in the rural areas.

### 4.2. Confirmatory Factor Analysis

The confirmatory factor analysis (CFA) was performed to verify the reliability and validity. As a result of the measurement model, Goodness-of-fit statistics for the structural model ($\chi2 = 711.297$, df = 398, $p < 0.000$, $\chi2/df = 1.787$, RMSEA = 0.085, CFI = 0.921, IFI = 0.922, TLI = 0.908) were judged to be overall excellent [72]. Factor loadings, significance probability of t−value, average variance extracted (AVE), and construct reliability (CR) were checked to check the convergent validity of the latent variables of the measurement model. The confidence coefficients (Cronbach's alpha) of factor loading were shown between 0.706 and 0.952, which was more significant than the 0.6 suggested by [73]. Moreover, AVE values and CE values were respectively constructed ranged from 0.707 to 0.855 and from 0.833 to 0.957. These values were all greater than that level of 0.5 and 0.7 suggested by [74]. In addition, correlation analysis was performed as shown in Table 1 to verify discriminant validity. As a result of Pearson's correlation analysis, all variables of NBSs, PPV, PPVB, PPN, ATEB, MH, WB, SA, and PRBI were $p < 0.01$, indicating a significant correlation association [75]. Thus, discriminant validity was confirmed.

**Table 1.** The measurement model and correlation.

| Construct and Scale Item | | Standardization Loading | NBSs | MN | WB | PPV | PPVB | PPN | ATEB | SA | PRBI |
|---|---|---|---|---|---|---|---|---|---|---|---|
| NBSs | NBSIO | 0.952 | 1 | | | | | | | | |
| | NBSOH | 0.823 | | | | | | | | | |
| MH | MH1 | 0.866 | 0.639 ** | 1 | | | | | | | |
| | MH2 | 0.878 | | | | | | | | | |
| | MH3 | 0.851 | | | | | | | | | |
| | MH4 | 0.902 | | | | | | | | | |
| WB | WB1 | 0.901 | 0.658 ** | 0.858 ** | 1 | | | | | | |
| | WB2 | 0.889 | | | | | | | | | |
| | WB3 | 0.915 | | | | | | | | | |
| | WB4 | 0.922 | | | | | | | | | |
| | WB5 | 0.888 | | | | | | | | | |
| PPV | PPV1 | 0.938 | 0.608 ** | 0.745 ** | 0.671 ** | 1 | | | | | |
| | PPV2 | 0.927 | | | | | | | | | |
| | PPV3 | 0.750 | | | | | | | | | |
| PPVB | PPVB1 | 0.868 | 0.618 ** | 0.743 ** | 0.703 ** | 0.888 ** | 1 | | | | |
| | PPVB2 | 0.843 | | | | | | | | | |
| | PPVB3 | 0.866 | | | | | | | | | |
| PPN | PPN1 | 0.882 | 0.633 ** | 0.762 ** | 0.695 ** | 0.821 ** | 0.895 ** | 1 | | | |
| | PPN2 | 0.780 | | | | | | | | | |
| | PPN3 | 0.870 | | | | | | | | | |
| ATEB | ATEB1 | 0.880 | 0.671 ** | 0.745 ** | 0.722 ** | 0.821 ** | 0.848 ** | 0.807 ** | 1 | | |
| | ATEB2 | 0.817 | | | | | | | | | |
| | ATEB3 | 0.908 | | | | | | | | | |
| | ATEB4 | 0.877 | | | | | | | | | |
| SA | SA1 | 0.947 | 0.693 ** | 0.755 ** | 0.779 ** | 0.691 ** | 0.710 ** | 0.715 ** | 0.751 ** | 1 | |
| | SA2 | 0.948 | | | | | | | | | |
| | SA3 | 0.878 | | | | | | | | | |
| PRBI | PRBI1 | 0.866 | 0.530 ** | 0.651 ** | 0.624 ** | 0.740 ** | 0.727 ** | 0.732 ** | 0.749 ** | 0.665 ** | 1 |
| | PRBI2 | 0.706 | | | | | | | | | |
| | PRBI3 | 0.889 | | | | | | | | | |
| | PRBI4 | 0.888 | | | | | | | | | |
| MEAN(SD) | | | 4.321 (0.531) | 4.252 (0.699) | 4.356 (0.707) | 4.242 (0.660) | 4.185 (0.692) | 4.079 (0.686) | 4.302 (0.658) | 4.476 (0.646) | 4.207 (0.650) |
| AVE (CR) | | | 0.792 (0.833) | 0.762 (0.929) | 0.816 (0.957) | 0.769 (0.907) | 0.738 (0.894) | 0.714 (0.882) | 0.759 (0.926) | 0.855 (0.947) | 0.707 (0.905) |

Note 1. NBSs = natural-based solutions, PPV = pro-environmental perceived value, PPVB = pro-environmental perceived belief, PPN = personal pro-environmental norms, ATEB = attitude toward environmental behavior, MH = mental health, WB = well-being, SA = satisfaction, PRBI = pro-environmental responsibility behavior intention. Note 2. Goodness-of-fit statistics for the structural model: $\chi2$ = 711.297, df = 398, $p < 0.000$, $\chi2/df$ = 1.787, RMSEA = 0.085, CFI = 0.921, IFI = 0.922, TLI = 0.908. Note 3. All factor loadings are significant at * $p < 0.5$, ** $p < 0.01$, *** $p < 0.001$.

### 4.3. Structural Equation Modeling

In this study, the PRBI of green hotels customers was investigated based on the VBN theory, NBSs theory, ART theory, and social exchange theory. The structural equation model (SEM) analysis was generated by using the maximum likelihood estimation method as an estimation method for both model and procedures' evaluation [72]. Goodness of Fit of Structural Model ($\chi2$ = 741.761, df = 410, $p < 0.000$, $\chi2/df$ = 1.809, RMSEA = 0.086, CFI = 0.917, IFI = 0.918, TLI = 0.908) was satisfactorily higher than the standard value. Moreover, SEM had shown high prediction power for PPV ($R^2$ = 0.861), ATEB ($R^2$ = 0.856), PPN ($R^2$ = 0.973), PPVB ($R^2$ = 0.921), MH ($R^2$ = 0.763), WB ($R^2$ = 0.837), SA ($R^2$ = 0.726), and PRBI ($R^2$ = 0.684) and t-values and standardized path coefficient were shown as the result in Table 2. The path estimates show that NBSs had a significantly positive direct effect on the PPV ($\beta$ = 0.928, $p < 0.001$), PPV had a significantly positive direct effect on the PPVB ($\beta$ = 0.963, $p < 0.001$), PPVB had a significantly positive direct effect on the PPN ($\beta$ = 0.989, $p < 0.001$), and PPN had a significantly positive direct effect on the PRBI ($\beta$ = 0.664, $p < 0.001$). Thus, H1, H2, H3, and H4 were supported. Moreover, NBSs had a significant positive on ATEP ($\beta$ = 0.925, $p < 0.001$), and ATEB had a significant positive on SA ($\beta$ = 0.332, $p < 0.01$); thus, H5 and H6 were supported. Moreover, the influence of

the NBSs on MH and WB was a significant positive MH ($\beta$ = 0.893, $p < 0.001$) and WB ($\beta$ = 0.915, $p < 0.001$); thus, H8 and H9 were supported. Additionally, MH had a significant positive effect on SA ($\beta$ = 0.389, $p < 0.01$), thus H10 was supported. However, WB had not been significantly associated with SA, and SA had not been positively significantly associated with PRBI; thus, H11 and H12 were not supported.

**Table 2.** The structural model results and hypotheses testing.

| Hypothesized Paths | Coefficients | t-Values |
|---|---|---|
| H1: NBSs → PPV | 0.928 | 9.394 *** |
| H2: PPV → PPVB | 0.963 | 12.570 *** |
| H3: PPVB → PPN | 0.989 | 9.794 *** |
| H4: PPN → PRBI | 0.664 | 5.971 *** |
| H5: NBSs → ATEB | 0.925 | 8.197 *** |
| H6: ATEB → SA | 0.332 | 2.333 * |
| H7: ATEP → PRBI | 0.280 | 1.858 |
| H8: NBSs → MH | 0.873 | 8.616 *** |
| H9: NBSs → WB | 0.915 | 8.895 *** |
| H10: MH → SA | 0.389 | 3.370 *** |
| H11: WB → SA | 0.186 | 1.858 |
| H12: SA → PRBN | 0.130 | 1.125 |
| Explained variable: $R^2$ (PPV) = 0.861 $R^2$ (ATEB) = 0.856 $R^2$ (PPVB) = 0.921 | $R^2$ (MH) = 0.763 $R^2$ (PPN) = 0.973 $R^2$ (WB) = 0.837 | $R^2$ (SA) = 0.726 $R^2$ (PRBI) = 0.684 |

Note 1. NBSs = natural-based solutions, PPV = pro-environmental perceived value, PPVB = pro-environmental perceived belief, PPN = personal pro-environmental norms, ATEB = attitude toward environmental behavior, MH = mental health, WB = well-being, SA = satisfaction, PRBI = pro-environmental responsibility behavior intention. Note 2. Goodness-of-fit statistics for the measurement model: $\chi 2$ = 741.761, df = 410, $p < 0.000$, $\chi 2/df$ = 1.809, RMSEA = 0.086, CFI = 0.917, IFI = 0.918, TLI = 0.908. * $p < 0.5$, *** $p < 0.001$.

### 4.4. The Moderating Effect on the Location (Urban, Rural)

In this study, an analysis was conducted to determine whether there was a moderating effect between NBSs and PRBI according to locations (urban, rural). If the customer had visited green hotels in the questionnaire, they were asked if they visited an urban or a rural location. As a result, out of 440 respondents, 255 respondents were the rural locations, and 188 respondents of an urban locations. The results of empirical comparisons are displayed in Table 3 and Figure 2. The goodness-of-fit statistics for the baseline model ($\chi 2$ = 1746.640, df = 820, $p < 0.000$, $\chi 2/df$ = 2.130, RMSEA = 0.92, CFI = 0.891, IFI = 0.896., TLI = 0.862) was showed an acceptable level for the data suitability. Subsequently, to examine the effectiveness of the conditioning effect, a constraint model was established by dividing the modulating variables into the urban group and the rural group, and the difference in $\chi^2$ between the free model and the constrained model was verified. There is a difference between the groups only if the $\Delta\chi^2$ of the baseline model (freely estimated) and the nested model (constrained to be equal) is 3.84 or more. As a result of the analysis, there was no difference between groups among PPV-PPVB, PPVB-PPN, ATEB-SA, WB-SA, and SA-PRBI. Therefore, H13b, H13c, H13g, H13k, and H13l were not supported.

**Table 3.** The result of the moderating effect.

| Paths | Urban Group (n = 188) | | Rural Group (n = 252) | | Baseline Model (Freely Estimated) | Nested Model (Constrained to Be Equal) | Chi-Square Different Test | Test Results |
|---|---|---|---|---|---|---|---|---|
| | Coefficients | t-Value | Coefficients | t-Value | | | | |
| H13a: NBSs-PPV | 0.874 | 6.279 *** | 0.924 | 5.640 *** | χ2(820) = 1746.640 | χ2(821) = 1752.877 | Δχ2(1) = 6.237, p > 0.01 | Support |
| H13b: PPV-PPVB | 0.928 | 7.759 *** | 0.969 | 9.158 *** | χ2(820) = 1746.640 | χ2(821) = 1746.683 | Δχ2(1) = 0.043, p > 0.01 | Not support |
| H13c: PPVB-PPN | 0.990 | 7.342 *** | 0.972 | 6.087 *** | χ2(820) = 1746.640 | χ2(821) = 1747.525 | Δχ2(1) = 0.855, p > 0.01 | Not support |
| H13d: PPN-PRBI | 0.057 | 0.453 | 0.848 | 3.728 *** | χ2(820) = 1746.640 | χ2(821) = 1754.304 | Δχ2(1) = 7.664, p > 0.01 | Support |
| H13e: NBSs-ATEB | 0.816 | 5.477 *** | 0.964 | 5.151 *** | χ2(820) = 1746.640 | χ2(821) = 1752.955 | Δχ2(1) = 6.315, p > 0.01 | Support |
| H13f: ATEB-PRBI | 0.964 | 5.027 *** | −0.189 | −0.841 | χ2(820) = 1746.640 | χ2(821) = 1756.143 | Δχ2(1) = 9.503, p > 0.01 | Support |
| H13g: ATEB-SA | 0.244 | 1.422 | 0.311 | 1.485 | χ2(820) = 1746.640 | χ2(821) = 1746.676 | Δχ2(1) = 0.036, p > 0.01 | Not support |
| H13h: NBSs-MH | 0.934 | 6.378 *** | 0.832 | 5.109 *** | χ2(820) = 1746.640 | χ2(821) = 1750.827 | Δχ2(1) = 4.187, p > 0.01 | Support |
| H13i: NBSs-WB | 0.999 | 7.199 *** | 0.880 | 5.230 *** | χ2(820) = 1746.640 | χ2(821) = 1750.298 | Δχ2(1) = 3.658, p > 0.01 | Support |
| H13j: MH-SA | −0.125 | −0.394 | 0.509 | 3.376 *** | χ2(820) = 1746.640 | χ2(821) = 1751.520 | Δχ2(1) = 4.880, p > 0.01 | Support |
| H13k: WB-SA | 0.795 | 2.096 | 0.042 | 0.239 | χ2(820) = 1746.640 | χ2(821) = 1749.052 | Δχ2(1) = 2.412, p > 0.01 | Not support |
| H13l: SA-PRBI | −0.129 | −0.744 | 0.208 | 1.535 | χ2(820) = 1746.640 | χ2(821) = 1748.920 | Δχ2(1) = 0.280, p > 0.01 | Not support |

Note 1. NBSs = natural-based solutions, PPV = pro-environmental perceived value, PPVB = pro-environmental perceived belief, PPN = personal pro-environmental norms, ATEB = attitude toward environmental behavior, MH = mental health, WB = well-being, SA = satisfaction, PRBI = pro-environmental responsibility behavior intention. Note 2. Goodness-of-fit statistics for the baseline model: χ2 = 1746.640, df = 820, p < 0.000, χ2/df = 2.130, RMSEA = 0.92, CFI = 0.891, IFI = 0.896, TLI = 0.862. *** p < 0.001.

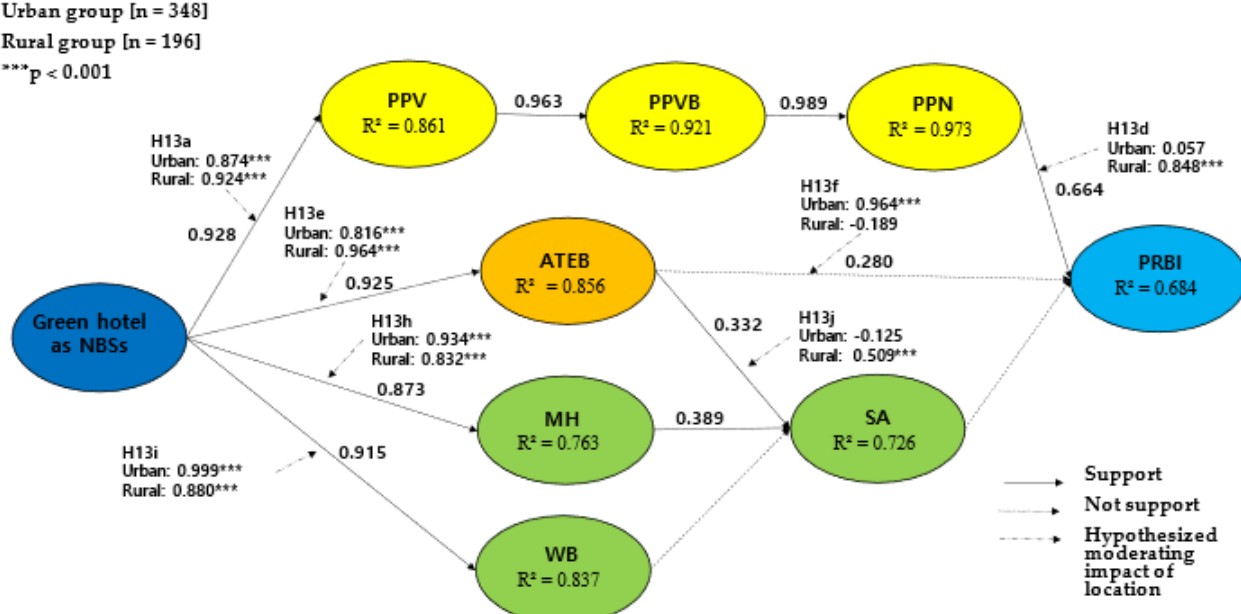

**Figure 2.** Structural equation model estimation and test for structural metric invariance.

## 5. Discussion

In recent years, many hotels around the world have contributed to the acceleration of global warming and continue to suffer natural disasters such as floods and typhoons due in part to the increase in plastic use in daily life (such as packaging containers and plastic cups) and medical supplies (such as face mask) [2–4]. Therefore, hotels are gradually implementing a Green Hotel as NBSs Policy for sustainable management [2,9], making

great efforts to raise the customer's PRBI. However, despite marketing activities to consider and create a sustainable environment, studies that can provide academic and practical implications for the personal relationship between their decision-making process and consumer behavior are lacking. For this reason, this study attempted to understand the PRBI of visitors to green hotels through VBN theory, SET, and ART. Moreover, the effect of green hotels as NBSs on PPV, PPVB, PPN, ATEB, MH, WB, SA, and PRBI was identified, and their relationship was verified. In addition, the difference between urban and rural green hotels was investigated by applying the moderating effect of location (urban, rural) in the relational model.

The summary of the results of this study is as follows. Green hotels as NBSs have a significant effect on PPV, BE, PEN, and PRBI and the result of these studies [3,15,34,35,37]. Other green hotels as NBSs have a significant effect on ATEB and SA, although ATEB has no significant effect on PRBI; therefore, previous studies [15,34] were partially supported. Green hotels and NBSs significantly affect MH and WB, which support previous studies [38–40]. However, WB has no significant effect on SA, not the same result as previous studies [76]. Moreover, SA has no significant effect on PRBI, not supporting previous studies [8,9].

Our result from the baseline model assessment and invariance test revealed that location (urban, rural) had a significant moderating role within the proposed conceptual framework. The paths from green hotels as NBSs to PPV (urban group: $\beta = 0.874$, $p < 0.01$ vs. rural group: $\beta = 0.964$, $p < 0.01$), from green hotels as NBSs to MH (urban group: $\beta = 0.934$, $p < 0.01$ vs. rural group: $\beta = 0.832$, $p < 0.01$), and from green hotels as NBSs to WB (urban group: $\beta = 0.909$, $p < 0.01$ vs. rural group: $\beta = 0.880$, $p > 0.01$), and ATEB to PRBI (urban group: $\beta = 0.964$, $p < 0.01$ vs. rural group: $\beta = -0.189$, $p > 0.05$) were significantly greater in the urban group than in the rural group. However, paths from PPN to PRBI (urban group: $\beta = 0.057$, $p > 0.05$ vs. rural group: $\beta = 0.848$, $p < 0.01$) and from ATEB to SA (urban group: $\beta = -0.125$, $p > 0.05$ vs. rural group: $\beta = 0.509$, $p < 0.01$) were significantly greater in the rural group than in the urban group.

### 5.1. Theoretical Implications

Based on the results of this study, theoretical and practical implications are deduced. The theoretic implications of this study are as follows. First, the PRBN of customers visiting green hotels was identified based on the VBN theory. Previous studies have identified general ecofriendly tourism behavior based on recognizing the importance of environmental behavior results of VBN theory [8,9,34,35,38–40]. In this study, the existing research was extended in that green hotels as NBSs identified PRBI through PPV, PPVB, and PPN. Second, green hotels are implementing NBSs policies to protect and preserve the environment to create a sustainable society and enterprises. Based on SET, it was determined that these corporate efforts influence customers' ATEB on PRBI. Third, the research framework has built NBSs into green hotels, and the researchers actively utilized the concepts of NBS theory, ART, and SET to construct this study. Our findings are theoretically significant in that they are the first empirical studies using SET and ART based on NBSs theory to determine PRBI. Last, in the study of NBSs, researchers need to understand this moderating nature of locations (urban, rural). They should actively utilize how they differ according to locations (urban, rural) as moderating when developing theory or conceptual frameworks related to NBS. Our findings have theoretical significance in that they are the first empirical studies to illuminate the interactions between NBSs and locations (urban and rural) that determine PRBI.

### 5.2. Practical Implication

The practical implications of this study are as follows. First, the green hotels' ecofriendly policy has a high impact on PPV and ATEB. These results show that caring for and protecting nature makes consumers feel more responsible for global warming, depletion of natural resources, and energy problems, and that hotel NBSs have a significant influence

on customer perceptions. Therefore, hotels should feel responsible for the environment and manage sustainable businesses. For example, Marriott International Inc announced in its 2021 SERVE 360 REPORT that it continues to produce positive effects to reduce waste by 45% by the end of 2030, reduce food waste by 50%, and water and carbon dioxide by 15% and 30%, respectively [77]. In addition, The Athenee Hotel, Bangkok's Luxury Collection Hotel, offers a Green Meeting Package of refreshments made of ecofriendly materials to reduce meeting and event waste. As a result, the Athenee Hotel achieved the world's first event sustainability management system standard (ISO: 20121) [78].

Second, although green hotel NBSs showed significant influence on PPV and ATEB, PPV had a significant effect on PRBN through the mediation of BE and PPN, while ATEB did not affect PRBI. In other words, it is believed this is because there is an intense psychological desire to copy others regardless of ATEB. Therefore, for green hotels' customers' PRBI to increase, it is necessary to recognize the customer's environment, such as PPV, PPVB, and PPN. As more people see the cause of COVID-19 being due to climate change caused by global warming [79], people's awareness of nature protection and sustainability is also increasing. Therefore, the demand for ecofriendly and alternative consumption is also increasing. For example, the French government has mandated the use of wood or natural materials for more than 50% of new public buildings built after 2022. Therefore, both hotel companies and government should increase the PRBI of green hotel customers to create a sustainable society.

Third, in the present research, we have indicated that MH derived from NBSs are fundamental concepts exerting a considerable influence on SA. However, WB had no significant effect on SA and MH had a significant effect on SA with lower values. Analysis of the hotel trends in 2022, including sustainable hotels [80]. Sustainable hotels are built using natural products while protecting nature, using trees or ecofriendly materials for construction, and restraint with accessories necessary for the hotel. In addition, it provides a pleasant atmosphere with relaxing music and a good aroma so that customers can heal comfortably in green hotels, promoting MH and WB and reducing mental fatigue. However, this has become a matter of course in green hotels. Therefore, to achieve high customer satisfaction, marketing is required one step ahead. For example, in 2022, another hotel trend, virtual reality, augmented reality, and sustainable hotels will be combined to help customers create and enjoy virtual spaces according to their needs without destroying nature.

Finally, because differences were analyzed between locations (urban, rural), it was found that an urban location had a more significant effect on the psychological wellness variable (MH, WB). In other words, green hotels located in urban areas are primarily near residential areas and can be easily accessed for rest and private time away from busy daily life. Thus, green hotels located in cities need to make a proper resting place for consumers. In addition, it was found that green hotels located in rural areas were more affected by socially required variables (PPV, ATEB) than those in urban locations. In other words, most customers of green hotels located in rural locations visited urban hotels because they wanted to find nature and heal. Because these customers want to find peace and stability in nature, they feel the importance of nature more and want to protect nature. Therefore, it is necessary to study how local green hotels can identify the needs of these customers, protect nature, and coexist with humans without destroying the environment. For example, the Banyan Tree restored Bang Tao Bay, which was little more than ruins, presenting it as Banyan Tree Phuket [81]. Phuket's Vantao Bay restoration project began of Banyan Tree's sustainable project, using natural materials for interior design and local unique architectural techniques. In addition, it preserves the community's environment where the Banyan Tree is located and supports social and economic development to help residents develop their abilities.

*5.3. Limitations*

The study used VBN, SET, and NBSs theories (theoretically certified for reliability and validity). Although it is significant in that it was intended to verify green hotel

users with various variables such as the characteristics of NBSs of green hotels and the psychological state of users, there are several limitations in this research process. First, as of December 2021, customers who visited green hotels in previous years' hotel stay experience were selected as samples, due to the specific situation of COVID-19; thus, visit surveys or interviews with sufficient explanation were not preceded. Instead, because we surveyed online, not only was the user's understanding of the survey low, but it was impossible to verify the change in attitude. It is hoped that this will lead to follow-up studies that can supplement future studies and solve limitations. Therefore, it is expected that standardized and generalized result analysis will be possible through regular and long-term surveys.

Second, the survey was targeted at visitors to ecofriendly hotels in Korea. Like Europe, citizens' awareness of ecofriendly hotels is still insufficient. Before the survey response, the characteristics of the green hotels were explained, although perceptions of hotels with ecofriendly marks and certification may be shallow. Thus, there may be gaps in survey responses through the experience of staying in the green hotels. In the aftermath of COVID-19, it will be necessary to explain to respondents directly with sufficient information using photos or pictures of ecofriendly hotels.

Third, the study analyzed the moderation effect of green hotels located in urban and rural areas, and a clear distinction between urban and rural areas may be unclear. The reason for this is that due to the geographical characteristics of Korea, the distinction between green hotels in the city center and the outskirts needs to be more apparent as the areas around lakes, rivers, seas, and mountains, where green hotels are located, are urbanized.

Fourth, this study did not consider hotel grades or global chains and local hotels. In future studies, it would be meaningful to identify the ecofriendly behavior characteristics of customers according to three-star, four-star, and five-star ratings and compare and analyze differences in the behavior of users according to the hotel star level. In addition, the comparative analysis of global chains and local hotels is also an essential part of predicting the ecofriendly behavior of users, so it is necessary to examine them in future studies.

**Author Contributions:** Conceptualization, S.Y. and T.K.; methodology, S.Y.; formal analysis, S.Y.; investigation, T.K.; resources, T.K.; data curation, T.K.; writing—original draft preparation, S.Y. and T.K.; writing—review and editing, S.Y. and T.K. All authors have read and agreed to the published version of the manuscript.

**Funding:** This research received no external funding.

**Conflicts of Interest:** Authors declare no conflict of interest.

## Appendix A. Measurement Items

Indoor of green hotels [8,9]
I quickly see green interior decorations and diverse living plants in the lobby area of green hotels
I quickly see various green items and light through glass windows in green hotels' restaurants.
I quickly see mixed flowers, trees, and potted plants in green hotels' coffee lounges.
Green space can easily be seen everywhere in green hotels.
Outdoor of green hotels [8,9]
Green hotels have easy access to the natural environment (i.e., mountains, forests, rivers, seas, lakes, natural parks).
The green hotels region has good weather (i.e., temperature, humidity, and precipitation).
The region surrounding green hotels has excellent and fresh quality air.
The region around green hotels are safe from natural disasters (i.e., earthquakes, typhoons, tsunamis, floods).
Pro-environmental perceived value [14]
Visiting the green hotels was worth the money paid.
Visiting the green hotels was a pleasant experience.
Visiting the green hotels was to improve self-esteem.
Pro-environmental perceives belief [14]
Staying in green hotels is a good idea.
Staying in green hotels makes you feel different from others.
I tend to think of Staying in the green hotels.
Personal environmental norm [14]
I feel obligated to visit green hotels.
Green hotels help the natural environment.
I use green hotels as my values and principles.
Attitude toward environmental behavior [14,40,54]
I think it has enough electricity, water, and trees in the Republic of Korea
I think recycling is necessary.

I think recycling is very important to conserve natural resources.
Green hotels try to preserve the natural environment.
Mental health [2,8,9,53]
Staying at green hotels plays an important role in relieving my mental stress and anxiety.
Staying at green hotels gave me confidence in my daily life.
Staying in green hotels is worth it because it makes me feel that I am a precious and important person.
Staying in green hotels is worth it because it helps turn all your anxiety and worries into confidence.
Well-being [2,8,9,53]
I feel healthy and happy during my stay in green hotels.
I feel emotionally secure during my stay at green hotels.
Green hotels play an important role in making my mind calm and peaceful.
Thanks to green hotels, I was able to relax comfortably.
Thanks to green hotels, I was able to refresh my mood.
Satisfaction [8,38,40]
Overall, I am satisfied with my experience at green hotels.
My decision to stay at green hotels was a wise one.
As a whole, I have enjoyed myself at green hotels as expected.
Pro-environmental responsibility behavior intention [43]
I tend to participate in the pro-environmental actions required by green hotels (use disposable items upon request, recycling, reusing linen, etc.).
I tend to try to be ecofriendly in green hotels.
I try to participate in the activities pursued by green hotels
I try to abide by the rules and regulations in green hotels.

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
