# Peer review of "Research Framework Built Natural-Based Solutions (NBSs) as Green Hotels"

_sustainability, doi:10.3390/su14074282_

Round 1

Reviewer 1 Report

Dear author/s,

The topic of the manuscript is interesting in the context of the sustainable development.

However from my point of view some aspects must be improved and clarified:

  1. I did not find the first 8 hypothesis
  2.  There are very nice statistical resulta, but the discussions are too poor.
  3. I was expecting that in the results section each stated hypothesis to be tested and the results presented

Good luck!

Author Response

Dear Editor and reviewers,

Thank you very much for reviewing our manuscript. We appreciate the opportunity you have afforded us to revise and resubmit. We found your suggestions to be thought-provoking and useful and have worked diligently to improve the manuscript as you suggested. Below, you will find our replies and responses to your constructive comments. Within the responses, red sections denote changes we made to the manuscript itself. Within the manuscript itself, changes are highlighted in red. We hope the changes made are satisfactory to you.

****************************************************************************************

Point 1: I did not find the first 8 hypothesis

Response 1: Thank you for the helpful comments. We added hypotheses that were not presented in the text.

Point 2: There are very nice statistical result, but the discussions are too poor. I was expecting that in the results section each stated hypothesis to be tested and the results presented.

Response 2: Thank you for your comment, and you raised good points. We have revised section 5. discussion to be clearer.

Reviewer 2 Report

Review

The paper is about the elaboration of a framework for natural-based solutions (NBSs) for green hotels.

It is an interesting paper on a neglected aspect related to sustainable tourism and hospitality literature. 

The abstract could be improved, trying to be more direct and maybe not just citing the different theories or concepts (We employed the value-belief-norm (VBN) theory , the attention restoration theory (ART), 10 and the social exchange theory (SET) to test its mediating effect on pro-environmental perceived 11 (PPV), pro-environmental perceived belief (PPVBE), personal pro-environmental norms (PPN), at-12 titude toward environmental behavior (ATEB), mental health(MH), well-being (WB), and satisfac-13 tion (SA) and the moderating effect of locations (urban, rural) between NBSs and pro-environmen-14 tal responsibility behavior intention (PRBI), and the moderating effect of locations.)

The abstract could be improved by a more plain style than it is at the beginning and at the end.

In the paper these theories and conceptual constructs should be more developed (either defined or characterized). Some definitional aspects could be put either in Annex to the paper or in a table. 

The review of the literature should situate better in relation to tourism, sustainable tourism and hospitality literature. The insertion of the research in this article is not well done as a contribution to the literature and the literature gap, beyond the SEM and the moderating effects of location. 

Section 2

It would be better to introduce the whole section on Conceptual framework. What is the general meaning and contribution of your model and conceptual framework?

In section 2, I would suggest a figure that illustrates the relationship between the variables and the mediating effects.

In the section 3 - methods

I would expect more details on the data collected and the measures of the variables. This section is short. We should understand the process better.

The section 4 of the results needs more clarity. The first paragraph (4.1) is not clear. A table could improve with a text explaining the data.

the same could be said for 4.2, 4.3 and 4.4.

Section 4.3 and 4.4 should be clearer and with a better presentation and interpretation of the results. 

figure 1 is difficult to read

The discussion of the results is interesting but lacks clarity.

Why is section 6 on patents empty?

Author Response

Dear Editor and reviewers,

Thank you very much for reviewing our manuscript. We appreciate the opportunity you have afforded us to revise and resubmit. We found your suggestions to be thought-provoking and useful and have worked diligently to improve the manuscript as you suggested. Below, you will find our replies and responses to your constructive comments. Within the responses, red sections denote changes we made to the manuscript itself. Within the manuscript itself, changes are highlighted in red. We hope the changes made are satisfactory to you.

****************************************************************************************

Point 1: The abstract could be improved, trying to be more direct and maybe not just citing the different theories or concepts (We employed the value-belief-norm (VBN) theory , the attention restoration theory (ART), 10 and the social exchange theory (SET) to test its mediating effect on pro-environmental perceived 11 (PPV), pro-environmental perceived belief (PPVBE), personal pro-environmental norms (PPN), attitude toward environmental behavior (ATEB), mental health(MH), well-being (WB), and satisfaction (SA) and the moderating effect of locations (urban, rural) between NBSs and pro-environmental responsibility behavior intention (PRBI), and the moderating effect of locations.)

The abstract could be improved by a more plain style than it is at the beginning and at the end

Response 1: Thank you for the helpful comments. Overall, the abstract part has been revised. Please see line 10-14 and 20-24 in page 1.

Point 2: In the paper these theories and conceptual constructs should be more developed (either defined or characterized). Some definitional aspects could be put either in Annex to the paper or in a table.

Response 2: Thank you for your comment. I made the table because a reviewer asked me to put theories and conceptual constructs in a table and put it in the paper. However, we have developed theories and conceptual constructs in 2. conceptual framework. we did not consider the table to be necessary and therefore did not include it in this study.

The table is presented below.

Theory

Author

Contents

VBN

Stern et al. (1999)

As the environmental behavior theory, Value-Belief-Norm (VBN), was developed.

Stern et al. (1999)

The values of the environment were divided into three categories: egoistic value, Altruistic value, and Ecological value.

Awareness of environmental pollution and ascription of responsibility for environmental issues that individuals can change the results through behavior were emphasized as important variables predicting eco-friendly behavior.

Stern (2000)

Value beliefs theory integrates the perspectives of Stern & Dietz (1994)'s Value Theory, Schwartz's Norm Activation Model (1977), and Dunlap & Van Liere (1978)'s New Environmental Paradigm.

Han et al. (2017)

Since it focuses on stricter structures and environmental areas than the norm activation model (NAM), it is used to explain tourists' participation in a wide range of green behaviors such as eco-friendly behavior and purchasing behavior for green products.

SET

Homans (1958)

Developed by Homans (1958),

In social exchange theory, when humans receive benefits such as compensation from the other party of the exchange, they form an exchange relationship that makes them feel obligated to repay it someday in the future.

Blau (1964)

Social exchange relations were explained by dividing them into economic and social exchanges.

A relationship that has a sense of obligation to bear social exchange theory someday when receiving benefits such as compensation from the other party.

Emerson (1962).

SET predicts that a person will leave a relationship when he/she perceives the costs of the relationship as outweighing the perceived benefits

NBS

Nature (2009)

NBSs are clearly described in the final report of the Horizon 2020 Expert Group

EC (2015)

NBSs are clearly described in the final report of the Horizon 2020 Expert Group

IUCN (2016)

The definition of NBS is not agreed upon worldwide but is recognized and used as a useful approach in various fields. NBSs cover all natural-based approaches, such as ecosystem-based adaptation and ecosystem-based mitigation

Faivre et al. (2017)

NBSs have been considered an optimized solution that is resilient, adaptable, resource-efficient, and locally adjustable for improving and maximizing the quality of life for urban residents

Han et al.(2020)

Green physical environments, such as hotel green spaces and existing outdoor natural environments, can be essential parts of NBS in the hotel context.

Point 3: The review of the literature should situate better in relation to tourism, sustainable tourism and hospitality literature. The insertion of the research in this article is not well done as a contribution to the literature and the literature gap, beyond the SEM and the moderating effects of location. 

Response 3: Thank you for your comment. We have modified the text by adding the part that needs you. Please see in page 7.

Point 4: It would be better to introduce the whole section on Conceptual framework. What is the general meaning and contribution of your model and conceptual framework

Response 4: Thank you for your comment. A conceptual framework has been added to the text. Please see line

345 and 360 in page 7.

Point 5: I would suggest a figure that illustrates the relationship between the variables and the mediating effects

Response 5: Thank you for your comment. This study did not include studies on mediating effects.

Point 7: I would expect more details on the data collected and the measures of the variables. This section is short. We should understand the process better.

Response 7: Thank you for your comment. I don't know which part of the collected data and the measurements of the variables should be further corrected. If you tell me more, I will edit it again.

Point 8: The section 4 of the results needs more clarity. The first paragraph (4.1) is not clear. A table could improve with a text explaining the data. the same could be said for 4.2, 4.3 and 4.4.

Response 8: Thabk you for the helpful comments. We have revised our section 4 in general.

Point 9: figure 1 is difficult to read

Response 9: Thank you for your comment. We modified Figure2 to make it look nice. Please see in page 13.

Point 11: The discussion of the results is interesting but lacks clarity.

Response 11: Thank you for your comment, and you raised good points. We have revised section 5. discussion to be clearer.

Point 12: Why is section 6 on patents empty?

Response 12: Thank you for your comment. Section 6 includes author contributions, funding, and conflicts of interest at the end of the paper. Let me know which part you need more and I'll fix it.

Reviewer 3 Report

Dear authors,

Thank you for sharing your research and manuscript on the Research framework built natural-based solutions (NBSs) as green hotels. In my opinion, the topic of the manuscript is interesting and both the hypothesis development and methodological sections are solid. There are some minor recommendations:

  • Introduction should contain the main purpose of the study
  • In the introduction the author/s should clearly state the scientific contribution - the gap in the literature that the authors are aiming to bridge should be identified, clear and highlighted.
  • Hypothesis – it is recommended that the hypotheses are presented in the form like H5 – H13. At this moment it is difficult to find H1, and I could identify H2, H3 and H4. On the p.4 the text starting from the line 189 up to line 196 is unclear

Good luck!

Author Response

Dear Editor and reviewers,

Thank you very much for reviewing our manuscript. We appreciate the opportunity you have afforded us to revise and resubmit. We found your suggestions to be thought-provoking and useful and have worked diligently to improve the manuscript as you suggested. Below, you will find our replies and responses to your constructive comments. Within the responses, red sections denote changes we made to the manuscript itself. Within the manuscript itself, changes are highlighted in red. We hope the changes made are satisfactory to you.

****************************************************************************************

Point 1: Introduction should contain the main purpose of the study. In the introduction the author/s should clearly state the scientific contribution - the gap in the literature that the authors are aiming to bridge should be identified, clear and highlighted.

Response 1: Thank you for your comment, and you raised good points. We have added it to the text to include the main purpose of the study. Please see line 98-106 in page 3.

Point 2: Hypothesis – it is recommended that the hypotheses are presented in the form like H5 – H13. At this moment it is difficult to find H1, and I could identify H2, H3 and H4. On the p.4 the text starting from the line 189 up to line 196 is unclear

Response 2: Thank you for the helpful comments. We added hypotheses that were not presented in the text. Moreover, The sentence that you said was unclear has been corrected.

Reviewer 4 Report

The study’s findings are systematically presented and inline with the study’s hypothesized associations between variables within the proposed theoretical framework and confirmed the moderating effect of location. The findings of the study have significant environmental ramifications, both theoretically and practically. The moderating influence of locations between NBS and PRBI, as proven in earlier research, is validated using NBSs, VBN, ART, and SET to explain the effect of the association between PPV, PPVB, PPN, ATEB, MH, WB, and SA on PRBI. Furthermore, the data presented here may motivate green hotels to assist in environmental problem avoidance. However these results were based on data collected from customers during the Covid-19 hence they may represent a varying perceptions as compared to a post Covid-era period. Thus, there may be gaps in survey responses through the experience of staying in the green hotels. In the future, if the aftermath of COVID-19, it will be necessary to explain to respondents directly with sufficient information using photos or pictures of eco-friendly hotels.  

Author Response

Dear Editor and reviewers,

Thank you very much for reviewing our manuscript. We appreciate the opportunity you have afforded us to revise and resubmit. We found your suggestions to be thought-provoking and useful and have worked diligently to improve the manuscript as you suggested. Below, you will find our replies and responses to your constructive comments. Within the responses, red sections denote changes we made to the manuscript itself. Within the manuscript itself, changes are highlighted in red. We hope the changes made are satisfactory to you.

****************************************************************************************

Point 1: The study’s findings are systematically presented and inline with the study’s hypothesized associations between variables within the proposed theoretical framework and confirmed the moderating effect of location. The findings of the study have significant environmental ramifications, both theoretically and practically. The moderating influence of locations between NBS and PRBI, as proven in earlier research, is validated using NBSs, VBN, ART, and SET to explain the effect of the association between PPV, PPVB, PPN, ATEB, MH, WB, and SA on PRBI. Furthermore, the data presented here may motivate green hotels to assist in environmental problem avoidance. However these results were based on data collected from customers during the Covid-19 hence they may represent a varying perceptions as compared to a post Covid-era period. Thus, there may be gaps in survey responses through the experience of staying in the green hotels. In the future, if the aftermath of COVID-19, it will be necessary to explain to respondents directly with sufficient information using photos or pictures of eco-friendly hotels. 

Response 1: Thank you for your comment. Based on the reviewers' comments, future research will receive another questionnaire to a post-Covid-era period. And then we will compare this research's questionnaire and the post-Covid-era period questionnaire. In addition, in future research, the questionnaire will be explained to respondents directly with sufficient information using photos or pictures of eco-friendly hotels.

Round 2

Reviewer 1 Report

Dear author/s,

thank you for the improved version of the manuscript.

Good luck!

Author Response

Point 1: thank you for the improved version of the manuscript.

Response 1: Thank you for reviewing.

Reviewer 2 Report

I consider the paper as much improved. There are three points to improve.

First, overall, to improve the clarity, style and grammar of the text.

Second, the research design, questions, hypotheses and methods could be better stated. The discussion is not always clear.

Third, for empirical research, the results are not always clearly presented.

I have joined the file with some comments to illustrate what I mean.

Author Response

Point 1: First, overall, to improve the clarity, style and grammar of the text.

Response 1: Thank you for your comment. We’ve improved the clarity, style and grammar of the text as a whole.

Point 2: Second, the research design, questions, hypotheses and methods could be better stated. The discussion is not always clear.

Response 2: Thank you for your comment. In general, it has been corrected and supplemented.

Point 3: Third, for empirical research, the results are not always clearly presented.

Response 3: Thank you for your comment. Efforts were made to present the results of the empirical study more clearly.
